



# DCMEX coordinated aircraft and ground observations: Microphysics, aerosol and dynamics during cumulonimbus development

Declan L. Finney[1,2], Alan M. Blyth[1,2], Martin Gallagher[3], Huihui Wu[3], Graeme Nott[4], Mike Biggerstaff[5], Richard G. Sonnenfeld[6], Martin Daily[1], Dan Walker[2,1], David Dufton[2], Keith Bower[3], Steven Böing[1], Thomas Choularton[3], Jonathan Crosier[3,7], James Groves[2], Paul R. Field[8,1], Hugh Coe[3,7], Benjamin J. Murray[1], Gary Lloyd[3,7], Nicholas A. Marsden[3,7], Michael Flynn[3], Kezhen Hu[3], Naveneeth M. Thamban[3], Paul I. Williams[3,7], James B. McQuaid[1], Joseph Robinson[1], Gordon Carrie[5], Robert Moore[9], Graydon Aulich[6], Ralph R. Burton[2], and Paul J. Connolly[3]

[1]Institute for Climate and Atmospheric Science, School of Earth and Environment, University of Leeds, Leeds, UK
[2]National Centre for Atmospheric Science, Leeds, UK
[3]Centre for Atmospheric Science, Department of Earth and Environmental Sciences, University of Manchester, Manchester, UK
[4]FAAM, Cranfield, UK
[5]School of Meteorology, University of Oklahoma, Norman, OK, USA
[6]New Mexico Institute of Mining and Technology, Socorro, NM, USA
[7]National Centre for Atmospheric Science, Manchester, UK
[8]Met Office, Exeter, UK
[9]Department of Geography and Environmental Sustainability, University of Oklahoma, Norman, OK, USA

**Correspondence:** Declan L. Finney (d.l.finney@leeds.ac.uk)

**Abstract.** Sensitivity of global temperature to rising $CO_2$ remains highly uncertain. One of the greatest sources of uncertainty arises from cloud feedbacks associated with deep convective anvils. For deep convective clouds, their growth and characteristics are substantially controlled by mixed-phase microphysical processes. However, there remain several questions about cloud microphysical processes, especially in deep, mixed-phase clouds. Meanwhile, the representation of these processes in global climate models is limited. As such, the Deep Convective Microphysics Experiment (DCMEX) has undertaken an in-situ aircraft and ground-based measurement campaign. The data, combined with operational satellite observations and modelling, will help establish new understanding from the smallest, cloud and aerosol particle scales through to the largest, cloud-system and climate scales. DCMEX is one of four projects in the UK Natural Environment Research Council, Uncertainty in climate sensitivity due to clouds, CloudSense programme. Along with other CloudSense projects, DCMEX will support progress in reducing the uncertainty in cloud feedbacks and equilibrium climate sensitivity. This paper lays out the underpinning dataset from the DCMEX summer 2022 field campaign. Its content describes the coordinated operation and technical details of the broad range of aerosol, cloud physics, radar, thermodynamics, dynamics, electric field and weather instruments deployed. In addition, an overview of the characteristics of campaign cases illustrates the complementary operational observations available, as well as demonstrating the breadth of the campaign cases observed.



# 1 Introduction

Equilibrium climate sensitivity is a fundamental metric for assessing the risks of CO2 emissions. Yet the plausible values of climate sensitivity have remained stubbornly uncertain for 40 years, with cloud feedbacks remaining a particularly uncertain component (Sherwood et al., 2020). The UK Natural Environment Research Council (NERC) has commissioned the Cloud-Sense programme to focus on this problem (https://cloudsense.ac.uk/). We present the observational campaign for one of the four CloudSense projects, the Deep Convective Microphysics Experiment (DCMEX).

Tropical high cloud, produced by deep convection, is an important cloud type when it comes to radiative effects and feedbacks (Bony et al., 2016; Hartmann et al., 2018; Gasparini et al., 2019). The IPCC Assessment Report 6 recently assessed there to be a negative feedback from tropical high cloud amount (e.g. cloud anvils) (Forster et al., 2021). This, however, came with low confidence that arises, in part, from the lack of understanding of the microphysical response to warming. Gettelman and Sherwood (2016), for example, pointed out that there is significant spread in cloud feedbacks across different GCMs due to uncertainties in the representation of microphysical processes.

Quantitatively explaining the development of the ice particle types and size distributions in convective clouds remains a fundamental problem. There are many questions surrounding the initial production of cloud ice on Ice Nucleating Particles (INP) (primary ice formation) (Kanji et al., 2017) and the development of high concentrations of cloud ice particles that dwarf the concentration of INPs (secondary ice production) (e.g. Cantrell and Heymsfield, 2005; Field et al., 2017). There are several candidate processes that might explain the unexpectedly high concentrations. The Hallett-Mossop (H-M) process of splinter production during riming (Hallett and Mossop, 1974) has been extensively investigated using aircraft measurements in cloud. Other, less studied processes include droplet shattering (Lauber et al., 2018; Lawson et al., 2022) and collision fragmentation (Yano and Phillips, 2011). Challenges that will be addressed using the DCMEX dataset include determining which process or processes can explain the observed distribution of cloud ice particles. If preliminary analysis of observations in DCMEX support previous results regarding the importance of the H-M process, another challenge will be to determine an improved parametrisation of the H-M process.

In July-August 2022, the DCMEX observation campaign was undertaken over the Magdalena Mountains, New Mexico. The aim was to carry out coordinated measurement of the aerosol, microphysics and dynamics of deep convective cloud formation. The Magdalena Mountains near Socorro, New Mexico provide ideal laboratory-like conditions for this study as isolated convective clouds form and grow over the mountains reliably during the North American summer (Dye et al., 1989). Our campaign built on microphysics-only measurements taken at the very same location in 1987 using the NCAR King Air aircraft (Blyth and Latham, 1993; Blyth et al., 1997; Blyth and Latham, 1997). Several important observations, which will guide analysis in DCMEX, arose from that early campaign:

- Primary ice particles, in concentrations consistent with the Cooper (1986) nucleation curve, were first observed when the in-cloud temperature reached about -10°C. Improved instrumentation in DCMEX should allow us to better detect primary ice particles, and relate them to concentrations of INP. A key step, since INP were not measured in the 1987 project.





- Clouds often contained supercooled raindrops that were observed prior to the formation of ice particles, despite the concentration of cloud drops being in excess of $700\,\mathrm{cm}^{-3}$.

- Clouds consisted of multiple thermals whose tops gradually ascended with time, until eventually there was a transition to a thunderstorm from cumulus congestus with tops at about -15°C (Raymond and Blyth, 1992). The sudden transition highlights a key feature for modelling electrification processes.

- There was evidence that the H-M process of splinter production during riming was responsible for the large concentration of ice particles. This result is consistent with subsequent research on the process. Improvements in cloud particle instrumentation, such as the ability to measure smaller particles and the reduction of ice shattering artifacts, offers the opportunity to increase our understanding and confidence in the H-M process.

- Finally, an interesting observation was made regarding cloud base. On the one occasion when the cloud base was much higher than usual due to lower humidity, the largest cloud droplets were too small to satisfy the criterion ($\mathrm{d} >= 24\mu\mathrm{m}$) for the operation of the H-M process (Mossop, 1978). A good understanding of such thresholds will enable more detailed parametrisations to be applied within models.

The DCMEX 2022 campaign described here has not only built upon the 1987 campaign through use of state-of-the-art cloud physics instruments, but also by coordinating observations of the whole aerosol-microphysics-dynamics-radiation system. This extensive dataset will be used to develop knowledge of microphysical processes, and improve microphysical parametrisations in models. Then, using these new tools and foundational understanding, the stage is set to target deep insights into convective cloud feedbacks that can help reduce uncertainty in equilibrium climate sensitivity.

A vast array of instruments were used for the campaign. The UK's BAe-146-301 Atmospheric Research Aircraft made measurements of cloud microphysics, aerosol and dynamics within the clouds whilst dual-Doppler radars and automated digital cameras monitored the cloud growth from nearby. Aerosol measurements, including of INPs, were collected on the aircraft and at the Langmuir Laboratory for Atmospheric Research on the summit of the Magdalena mountain range (33.98N, 107.18W). Within the DCMEX project, these data will be analysed in combination with satellite radiation products from the Geostationary Operational Environmental Satellite (GOES) R Series and the Clouds and the Earth's Radiant Energy System (CERES). Meanwhile, support of modelling activities will focus on the recently developed Cloud-AeroSol Interacting Microphysics (CASIM) module that can be used within the Met Office Unified Model (Miltenberger et al., 2018a, b; Hawker et al., 2021; Field et al., 2023). Altogether, the dataset will enable: 1) the development and testing of the microphysics schemes applied in global climate models, and 2) increased understanding of deep convective processes that impact cloud radiative effects and feedbacks. These two components will support the overarching goal of DCMEX to reduce climate sensitivity uncertainty.

## 2 Flight and ground-based operations

In total there were 19 flights over the course of the 24 days between 16th July and 8th August 2022. Every flight involved takeoff from Albuquerque International Sunport between 15:00 and 16:15 UTC (9 to 10:15 am, local time, i.e. Mountain

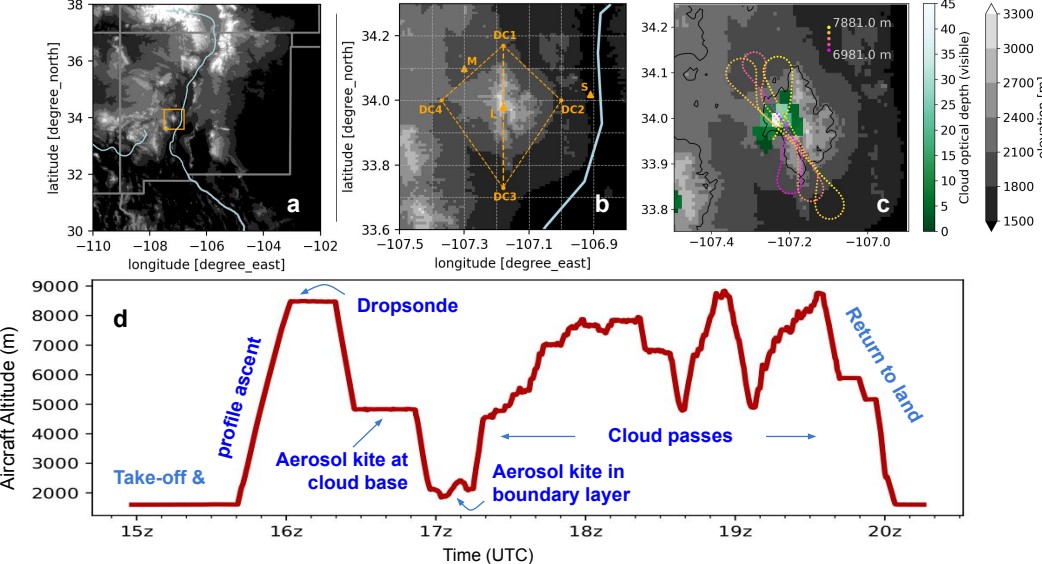

**Figure 1.** The main study region and representative flight paths. a) The DCMEX study region (box) in the context of the New Mexico, USA terrain. State borders are shown in grey. Rivers, including the Rio Grande in New Mexico, are shown in light blue. b) Core flight coordinates and locations of instruments. DC1-DC4 polygon shows the kite path that was used for aerosol runs, the DC1-DC3 line shows the nominal path for cloud passes, though there was substantial deviation from this. Letter L marks Langmuir Laboratory, S marks Socorro airport, and M marks Magdalena airport. The airports hosted the radars and cameras, and the Laboratory hosted weather, aerosol and electric field instruments. C) Flight track locations/altitudes between 17:45z and 18:15z on the 22nd July flight. This is plotted over the GOES cloud optical depth observation at 18:02z. The cloud optical depth field was corrected for parallax shift on a pixel-by-pixel basis using GOES cloud top height product (Ayala et al., 2023), the result was then regridded to 0.1° regular grid for plotting. Black contour shows 2250m terrain height. d) Flight altitude and activities from 22nd July. The 22nd flight provides a illustration of the general flight characteristics.

Daylight Time). Flight durations varied between approximately 3 - 4.5 hours (Table 1). Each flight involved a profile ascent to 8-9 km followed by deployment of a dropsonde in the vicinity of the Magdalena Mountains. Over the course of the rest of the flight there were a mixture of cloud passes and aerosol runs, depending on conditions. Aerosol runs were generally conducted first, partly to characterise the airmass that the clouds formed within, and partly to allow for rapid response to convective

initiation once it started. Figure 1 shows the key waypoints used for the majority of runs during flights. In addition, a few runs were made around the San Mateo Mountains to the southwest when clouds were not present over the Magdalena Mountains. Figure 1 illustrates the flight stages described above, as well as example cloud passes undertaken during the campaign.

Basic details regarding the cloud and aerosol runs are provided in Table 1. Aerosol runs around the base of the mountains took the form of a kite with runs between waypoints designated DC1 (34.17N, 107.18W), DC2 (34.00N, 107.00W), DC3

(33.73N, 107.18W), and DC4 (34.00N, 107.37W) (Figure 1). The kite was flown either clockwise or anti-clockwise, conditions



depending, and was used to sample aerosols, including INP, and thermodynamics within the boundary layer inflow. As well as low-level, terrain-following runs, aerosol kite runs were also carried out close to cloud base height, and at higher altitudes in relatively clean free-tropospheric air.

Cloud passes generally aimed to sample developing congestus clouds at various heights from close to cloud base up to about the -20°C isotherm. Two approaches were used as deemed appropriate by the mission scientist: 1) To sample congestus turrets multiple times ∼500ft below cloud top as they grew over the course of the flight, or 2) repeated sampling between -3 and -10°C (the H-M zone). The first approach targeted mainly initial ice formation where it was known there was no influence from falling ice. The second approach focused on forming a time series of ice formation within the mixed-phase region especially known for secondary ice formation. Secondary ice due to the HM process could also be sampled in the first approach due to multiple thermals and the time taken to ascend to low temperatures. When sensible to do so, cloud passes followed the north-south line between DC1 and DC3 (Figure 1), as this followed the mountain ridge and broadly aligned parallel to the prevailing wind flow. As intense cumulonimbus clouds developed it was not always possible to take this path, and alternatives were developed as required and based on the conditions at the time.

To the east and northwest of the Magdalena Mountains are the Socorro and Magdalena airports, respectively. These were used as the locations for the radars and automated digital cameras. Together they provided a more comprehensive overall view of the cloud than the aircraft could provide alone, as well as monitoring the cloud continuously both before and after the aircraft was sampling. In addition to each instrument's unique perspective, the coincident measurements of different instruments will allow more detailed description of cloud growth, e.g. through better constrained estimates of turret ascent rates.

Whilst the aircraft measured boundary layer aerosol in each flight, a static continuous measurement at the surface is a beneficial addition. Therefore, aerosol and INP samples were collected at Langmuir Laboratory for Atmospheric Research on top of the Magdalena Mountains. Automatic weather stations were also installed to provide continuous local surface weather. The Langmuir Laboratory has been extensively used for storm electrification measurements (Edens et al., 2019; Jensen et al., 2021), and provided live electric field measurements that were key, in combination with live radar, for avoiding first lightning stroke as storms developed.

A lightning hit was a high risk for aircraft instruments and the aircraft. Whilst not a safety risk in flight, a lightning strike requires extensive aircraft maintenance post-flight that can potentially rule out flying for the remainder of a research campaign. Pilots have access to the aircraft's radar and have their protocols for determining when it becomes too dangerous to fly in a cloud. This process and decision took precedent over all else when flight paths were determined. To further reduce risks, additional information sources were assessed to determine when a cloud approached a high probability of producing a lightning strike. These sources were the electric field measurements at the Langmuir Laboratory and the radar measurements from Socorro and Magdalena airports. Lead scientists stationed at the laboratory and radars communicated their near-realtime observations through the aircraft internet-connected, text-based CHAT system. This was constantly monitored by the mission scientists onboard, and the DCMEX principal investigator, who was based at the control centre in Albuquerque. Blyth took responsibility for assessing information on radar reflectivity, electric fields and observations from the aircraft scientists to pro-



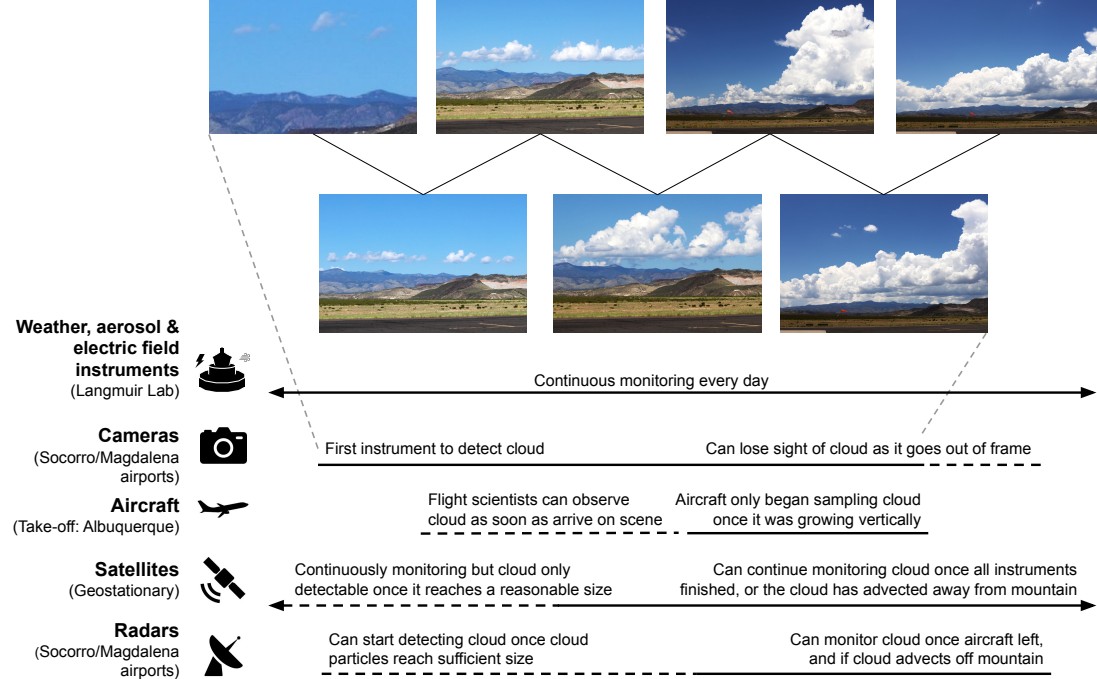

**Figure 2.** Indicative stage of cloud growth at which different instruments made observations and detected the cloud. Supplementary Tables 1-3 provides details of instrument operation for each day of the campaign.

vide a ground-based recommendation to pilots of when to avoid flying in clouds. Anecdotally, decisions from the control centre timed very well with the pilots' own decisions. This increased confidence that the process was robust.

The above measurements complement weather station, satellite and sonde releases already in operation across New Mexico. In particular, the GOES/CERES satellite imagery will prove invaluable when relating microphysical processes to the radiative properties of the cumulonimbus anvils. Figure 2 illustrates the spatial and temporal relationships between instruments, and

130 Supplementary Tables 1-3 lists details of the instrument operation across the campaign.

Flight days were mainly decided on the preceding day. Decisions were partly informed by national and local operational forecast tools, including the High Resolution Rapid Refresh forecast model produced by the National Oceanic and Atmospheric Administration of the USA. In addition, three bespoke high-resolution model forecasts were produced daily during the DCMEX campaign. The models used were the UK Met Office Unified Model (configurations: RA2m and RAL3) and the Weather

Research and Forecasting model. These models were able to clearly simulate cumulonimbus development, and on the whole provided robust forecasts in line with the ebb and flow of the convective activity during the campaign.

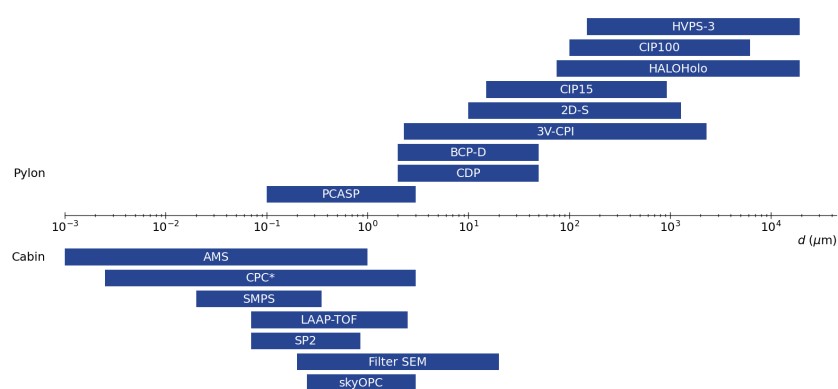

**Figure 3.** Nominal measurement size ranges of DCMEX particle instruments installed in the aircraft cabin and on under-wing pylons. The CPC (asterisk) counts particles within the indicated size range. All other instruments provide size distributions of detected particles. The effect of inlets on the ranges of the in-cabin instruments is included.

## 3 Instrumentation

Many different UK and US research teams came together to provide a coordination of instruments for this campaign. Below is a list of the key instruments operated to produce data to address DCMEX objectives.

### 3.1 FAAM BAe-146 aircraft

The FAAM BAe-146 aircraft is owned by UK Research and Innovation and NERC. It is managed through the National Centre for Atmospheric Science to provide an aircraft measurement platform for use by the UK atmospheric research community on campaigns throughout the world. A bespoke configuration of instruments, concentrating on measurements of dynamics, thermodynamics, aerosols, and cloud particles, were installed on the aircraft for DCMEX. Most aerosol instruments were installed in the cabin behind various inlets while cloud spectrometer and imaging probes were installed on pylons under each wing. During sampling runs the aircraft flies at a constant 200 kts ($102.8\ \mathrm{ms^{-1}}$) indicated air speed. Thus true air speed increases with altitude (with a corresponding decrease in the spatial resolution of measurements).

Figure 3 summarises the particle size detection range of the aerosol and cloud instruments aboard the aircraft. They cover the important sizes required for the research, spanning from the submicron to millimetre and centimetre range. An overview of each instrument and its operation is provided in the following sections.



### 3.1.1 Aerosol instruments

The aircraft used for DCMEX was equipped with a series of online aerosol instruments (determining aerosol loadings, chemical composition and size distributions) and offline characterisation of INP. The characteristics of aerosol properties, ingested into the base of the cloud, are of interest to interpret the size distribution of cloud droplets at cloud base and the distribution of

primary ice particles (forming later). It also provides a signature of the air masses that influence the clouds, providing a link to synoptic scales. It is not only the low-level, boundary layer aerosol particles that are of interest. There is the possibility of entraining INP and cloud condensation nuclei (CCN) into the cloud at higher levels. Furthermore, aerosols at such higher levels may have been processed through previous clouds and left in detrained cloud layers or anvils before re-entering the clouds of interest.

In this study, a Counterflow Virtual Impactor (CVI) inlet was used. The working principles of the CVI inlet are described in detail by Shingler et al. (2012). The CVI inlet with counterflow on is used to sample residue particles of cloud droplets. It only allows cloud droplets larger than the cut size coming into the inlet, and obtains cloud residue particles by using dry and particle free carrier air to evaporate the cloud water. During the campaign, the droplet cut size used was approximately 6.5 $\mu$m (aerodynamic diameter). The remaining cloud droplet residues can then be characterised by some online aerosol instruments

behind the CVI inlet. Concentrations measured behind the CVI inlet have to be divided by an enhancement factor, which can be calculated based on the methods in Shingler et al. (2012). Furthermore, when the counterflow is off, the CVI inlet allows total air coming into the CVI inlet and can be used to sample ambient aerosols out of cloud.

The principles and operation of the main aerosol instrumentation are listed below:

– **Aerosol Mass Spectrometer (AMS)**. A compact time-of-flight aerosol mass spectrometer (C-TOF-AMS) was employed

to measure the chemical composition of non-refractory submicron aerosols (i.e., organic aerosol (OA), sulfate, nitrate and ammonium), enabling chemical characterization across a spectrum of ion mass-to-charge ($m/z$) ratios from 10 to 500 Drewnick et al. (2005). Previous aircraft work has provided detailed operation of the AMS, including calibration and correction factors (i.e, Morgan et al. (2010)). Briefly, the aerodynamic lens inlet system of the AMS focuses the entering particles into a narrow beam, which is tailored to sample submicron aerosols. After the inlet system, particles come into

the particle-sizing chamber, which is gradually evacuated to lower pressures. The strong vacuum in the chamber removes the majority of gases, as the particle beam passes through it. Subsequently, the particles undergo flash vaporization upon encountering a resistively heated porous tungsten surface. The resultant gaseous molecules are then subjected to an ionization step with a 70-eV electron beam emitted from a tungsten filament. The fragment ions after ionization step are then examined with a Time-of-Flight mass spectrometer (TOF-MS). The transmission of particles beam to

the TOF-MS is controlled by a "chopper", the position of which decides the data record modes. When the chopper is fully open, the MS (mass spectrum) mode is collected to record average mass spectra for the ensemble of particles. For MS mode analysis, background mass spectra need to be collected by fully closing the chopper. When the chopper is placed in a "chopped" position, The P-TOF (Particle Time-of-Flight) mode is collected to record averaged mass size distribution data for the ensemble of particles. In this study, we employed an improved particle size measurement





module, the efficient Particle Time of Flight (e-PTOF), which has a better signal-to-noise ratio with a $\sim 50\%$ particle
throughput. AMS calibration involved the utilization of monodisperse particles of ammonium nitrate and ammonium
sulfate. The AMS data underwent processing through the SQUIRREL (SeQUential Igor data RetRiEvaL, v. 1.65C)
TOF-AMS software package (https://cires1.colorado.edu/jimenez-group/ToFAMSResources/ToFSoftware/, visited: 25
Aug, 2023). To achieve better accuracy, we employed an algorithm introduced by Middlebrook et al. (2012) to correct
data with a time and composition dependent collection efficiency.

– **Laser Ablation Aerosol Particle Time-of-Flight (LAAP-TOF) mass spectrometer**. The LAAP-TOF (AeroMegt
GmbH) was used in this study to provide real-time identification of chemical composition of individual aerosol par-
ticles. The system of the LAAP-TOF has been described in detail by Marsden et al. (2016, 2018). In brief, aerosols
are sampled via an aerodynamic lens inlet, focusing the incoming particles into a narrow but divergent beam. In this
study, the transmission of particles in the aerodynamic diameter range of 0.07–2.5 $\mu$m is similar to the high-pressure
lens described by Williams et al. (2013). Afterwards, they pass through an optical detection chamber based on a multi-
wavelength laser system (iFLEX-Viper Multi-Line Laser Engine, Qoptiq). The optical particle detection stage is used
for temporal alignment of the excimer laser pulse with the presence of a particle in the ion source range. The option to
measure particle optical size was not used in this study. Detecting particles smaller than 500 nm and larger than 2.5 $\mu$m
is challenging due to the diminished light scattering exhibited by the smaller particles and the increased particle beam
divergence associated with the larger particles. On the FAAM aircraft the inlet system further reduces the sampling ef-
ficiency of the LAAP-TOF for larger particles. Once a single particle is detected successively by the detection lasers,
an excimer laser pulse ($\lambda$ = 193 nm, 8 ns pulse) is triggered for a one-step desorption/ionization of the particle. The
resulting cations and anions are analysed by a bipolar time-of-flight mass spectrometer (Tofwerks AG). Ion arrival times
at the multichannel plate detectors, one for each ion mode, are recorded by a dual channel 14 bit analogue to digital
converter (model ADQ214, SP Devices) with a bin width of 2.5 ns. The general data analysis for single particle spectral
information was undertaken with the LAAP-TOF data analysis Igor software (Version 1.0.0, AeroMegt GmbH).

– **Passive Cavity Aerosol Spectrometer Probe (PCASP)**. A PCASP with SPP-200 electronics was operated in a wing-
mounted canister. This instrument provides aggregated 5 Hz particle numbers in 30 size bins across a nominal diameter
range 0.1–3 $\mu$m. The smallest bin is discarded due to an undefined lower boundary and bins are merged at the gain-stage
crossover points as described by Ryder et al. (2013). Particles are binned according to the strength of the photovoltage
generated by HeNe laser light scattered by each particle. Laboratory calibrations both before and after the campaign are
used to convert photovoltages into scattering cross-sections for each bin (Rosenberg et al., 2012). These calibrations are
provided in separate files alongside the data files to be applied by the data user. With knowledge of the aerosols being
sampled, that is particle shape and complex refractive index, the scattering cross-sections can be converted into particle
diameters. This information must be determined through other means and applied by the users to obtain calibrated
particle sizes and thus size distributions and any required derived parameters. The volumetric flow rate, used to calculate



particle concentrations, was calibrated in the laboratory using either a Gilibrator 2 [Sensidyne LP] low-flow wet cell or, more recently, a Gilibrator 3 dry cell calibrator.

– **Scanning Mobility Particle Sizer (SMPS)** and **GRIMM sky Optical Particle Counter (skyOPC)**. Aerosol number size distributions were measured in this work via combination of a SMPS (Wang and Flagan, 1990), a GRIMM Aerosol Technik 1.129 skyOPC (Grimm and Eatough, 2009), and the PCASP described above. The SMPS collected samples from the same inlet as the AMS and assessed distributions of dry particle mobility diameter. Diameters were categorized into 26 or 31 logarithmically spaced bins within the range of 20 to 350 nm. To achieve this, a low-pressure, water-based

condensation particle counter (WCPC model 3786-LP) was linked to a TSI 3081 differential mobility analyzer. The SMPS scans through a voltage range and is able to produce a full-size distribution of aerosol particles (20 – 350 nm) approximately once per minute. Given the time resolution, SMPS data are only available in straight and level runs and without rapid aerosol concentration changes. The SMPS data can be inverted using the inversion algorithms developed by Zhou (2001). The skyOPC instrument uses the principle of light scattering intensity to measure the size of a particle

at 1 Hz. Here, the skyOPC was operated in the fast mode for smaller sizes, covering a nominal diameter range of 0.25 to 3 $\mu$m.

– **Single Particle Soot Photometer (SP2)**. The refractory black carbon (hereafter referred to as BC) was characterized using an SP2 (Droplet Measurement Technologies, Boulder, CO, USA). The instrument setup, operation and data interpretation procedures can be found elsewhere (McMeeking et al., 2010; Liu et al., 2010). The SP2 detects particles

with an equivalent spherical diameter in the range of 70 - 850 nm. It can determine the BC mass within those particles and hence the BC mixing state. When there is an aerosol particle containing absorbing BC material passing through the laser beam, the BC-containing particle absorbs the laser and heats up. When the BC-containing particle reaches its boiling temperature, it will incandesce and emit visible light. Two detectors in the SP2 will capture the signal and identify the absorbing particle as an incandescing BC-containing particle. The SP2 incandescence signal is proportional to the

mass of refractory BC present in the particle, regardless of mixing state. The SP2 incandescence signal was calibrated using Aquadag black carbon particle standards (Aqueous Deflocculated Acheson Graphite, manufactured by Acheson Inc., USA), including use of the correction factor (0.75) recommended by Laborde et al. (2012). The mass can be then converted to a spherical-equivalent BC core diameter with an assumed BC density of 1.8 g cm$^{-3}$.

– **Teflon and polycarbonate filters**. Aerosol for offline INP and compositional analysis were collected in parallel onto

a pair of filters - polycarbonate track-etched membranes with 0.4 $\mu$m pore diameter (Whatman-Nuclepore 10417112) and Polytetrafluoroethylene (PTFE) membranes with 1.2 $\mu$m effective pore diameter (Sartorius type 11806) - from air sampled by the dual aircraft inlet. Sampling runs typically lasted 10-20 minutes and sampled volumes of air ranging between 87 – 987 L depending on altitude, filter pore size and filter support type, as calculated using air flow rates for each channel determined using an in-line flowmeter and datalogger. A full characterization of this system is given in

Sanchez-Marroquin et al. (2019) and examples of its previous use for sampling INP are given in Price et al. (2018) and Sanchez-Marroquin et al. (2021). An adaptation from previous campaigns was the use of wider PTFE filter pore sizes and





less restrictive support meshes to enable higher flow rates and thus enabling sampling at higher altitudes than in previous campaigns (up to 6,500 m). Polycarbonate filters were divided and used for offline scanning electron microscopy analysis (SEM) and INP analysis, while PTFE filters were used for INP analysis only. Blank filters were taken on each flight to

establish the limit of detection for INP concentrations, where a pair of filters (a polycarbonate and a PTFE each) were prepared and loaded into the sampling system as normal but only exposed to ambient air for around one second. INP analysis by droplet freezing assays (DFAs) combined with total air flow were used to determine INP concentrations per litre of air for each sampling run. A temporary laboratory for DFAs and clean handling of filters was established in Albuquerque which allowed the PTFE filters to be analysed for INP within 24-48 hours of collection. The polycarbonate

filters were stored in airtight filter cassettes, transported back to University of Leeds and stored at -20 °C for DFA and SEM analysis. The hydrophobicity of PTFE filters enables use of the 'drop-on' DFA technique where droplets of pure water are placed directly on the exposed filter placed on a cooling stage (Price et al., 2018). Polycarbonate filters were analysed for INP using the 'wash-off' method, where the filter is placed in pure water to create a suspension that is subsequently pipetted onto a clean substrate on a cooling stage (Whale et al., 2015). Using the 'drop-on' DFA technique

with PTFE filters enabled a higher sensitivity sampling of INPs (0.01 - 10 $\mathrm{L}^{-1}$), compared to the wash-off method (1 - 100 $\mathrm{L}^{-1}$) as the particles on the filter are not diluted by entering a suspension. Therefore, in combination with the higher air flow rates due to the larger pore size used, the 'warmer' end of the INP spectrum for a single sampling run is captured by analysis of PTFE filters, while the 'colder' end is captured via the polycarbonate filters. A polycarbonate and PTFE filter was obtained for almost all aerosol run heights listed in Table 1. The only exceptions were that 2 PTFE filters were

collected at each height on the 19th and 20th July, i.e. no polycarbonate filters were collected on those days. This was to ensure both filter channels were providing equivalent samples. Selected filters were analysed by SEM (Tescan VEGA3 XM fitted with an X-max 150 SDD energy-dispersive X-ray spectroscopy) at the University of Leeds to determine the morphological and elemental composition of particles above 0.2 $\mu$m collected on the polycarbonate filters. This method, outlined in Sanchez-Marroquin et al. (2019), served to characterise the size distribution and size-resolved composition of

the collected aerosol using automated particle scanning. Classification software (Aztec 3.3, Oxford Instruments) enabled thousands of particles per filter to be individually scanned on each filter and automatically classified into compositional classes such as mineral dust, carbonaceous and sulphate-rich particles, according to size.

During the campaign, when the aircraft was flying in clouds, cloud residues were sampled downstream of a CVI inlet with counterflow on. Cloud residue number concentrations were measured with a butanol-based condensation particle counter (CPC,

TSI model 3010), which detects aerosol particles larger than 10 nm at 1-Hz resolution. Cloud residue number size distributions were measured by the GRIMM skyOPC. The chemical composition/mixing state of cloud residue can be analysed by the LAAP-TOF, AMS and SP2. When the aircraft was flying out of clouds, the inboard instruments including the LAAP-TOF, the butanol-based CPC, and GRIMM skyOPC sampled ambient air via the CVI inlet with the counterflow off. The inboard aerosol instruments including the AMS, SP2, SMPS sampled ambient air via stainless steel tubing from a modified Rosemount inlet,

which has sampling efficiencies close to unity for submicron particles (Trembath, 2013).



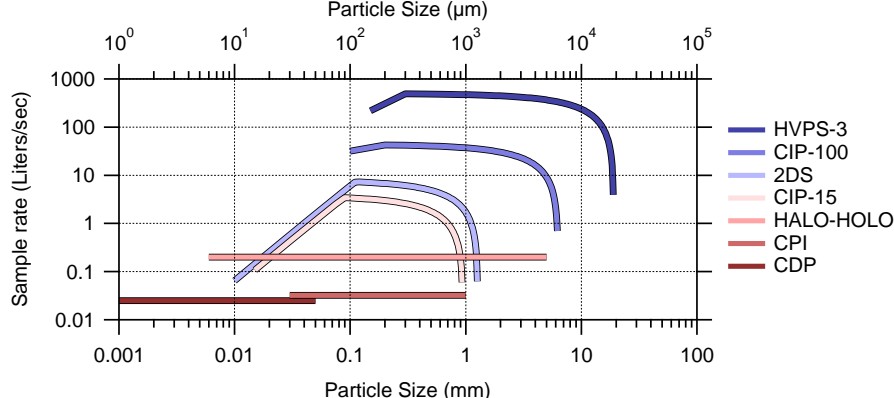

**Figure 4.** Nominal sampling rate of the various cloud particle detectors operated on FAAM during the DCMEX campaign assuming an airspeed of 100 m s$^{-1}$. The size dependence in the sampling rate for the Optical Array Probes (HVPS-3, CIP-100, 2DS and CIP-15) is a result of a) the post-processing which rejects partially imaged particles, and b) the size dependence of the Depth-of-Field of the imaging systems (Knollenberg, 1970). The sample volumes assume particles are spherical, and do not include the effects of dead-time and coincidence, which vary with ambient concentration.

Combined, the instrumentation described above characterises the chemical composition and size distributions of aerosols. In addition, the potential for primary cloud ice formation can be established through INP measurements.

### 3.1.2 Cloud physics instruments

The purpose of making aircraft cloud physics measurements in DCMEX was to provide information regarding the temporal
and spatial distribution of cloud particles as the clouds developed. The instruments together provide coverage of the full range of cloud particle sizes and properties including quantification of concentrations and ice mass as a function of ice crystal habit. In addition, they enable examination of fine morphological details to probe primary and secondary ice production (SIP) processes. Specifically, the data will be used to determine the properties of the primary and secondary ice particles, as well as where precipitation particles first form and how they develop. A thorough review including instruments used here was carried
out by Baumgardner et al. (2017).

Figure 3 illustrates the detectable particle size range of each instrument discussed in this section, while figure 4 presents how the sampling efficiency varies by particle size for the cloud physics instruments. This is particularly important to consider when analysing high-frequency variations in the data.

– **Two Dimensional (Stereo) probe (2D-S)**. The 2D-S instrument is the key cloud instrument for determining ice particle
concentrations as a function of size and habit. It consists of high-speed, dual 128-photodiode linear array channels (or-thogonal to each other and the direction of flight) and electronics to produce shadow-graph 2D stereo images of particles covering the nominal size range 10-1280 $\mu$m, with a resolution of 10 $\mu$m (Lawson et al., 2006). The stereo imaging capability allows for minimization of errors associated with image depth of field, sample volume uncertainties, and time





response errors. Images can be captured at rates up to 74 frames per second depending on available data transmission rates. The sample volume of the instrument is approximately 16 L at an airspeed of 100 m s-1. The instrument was also fitted with Korolev anti-shatter tips (Korolev et al., 2011; Lawson, 2011) to minimise particle shattering artefacts. Analysis of 2D-S particle inter-arrival time histograms is used to identify and remove potential shattered particles (Field et al., 2006). Discrimination between spherical and irregular particles is determined for particles typically greater than 50-100 $\mu$m in size using a circularity criterion (Crosier et al., 2011; Lloyd et al., 2020). The particle shape categories generated include low irregular (LI, with a defined shape factor between 1 and 1.2), indicating liquid droplets, or newly frozen liquid droplets that maintain a near spherical shape; medium irregular (MI, shape factor between 1.2 and 1.4), for increasingly irregular particles, likely indicative of ice; and high irregular (HI, shape factor > 1.4), indicating ice particles. Particles comprised of fewer pixels than a set threshold number (e.g. 20 pixels) are assigned to an "Unclassified" shape category. The high sampling rate and resolution of the 2D-S allows possible identification of regions where ice crystals are at their embryonic stage of formation and SIP mechanisms may be occurring (Lawson et al., 2006). However, in high cloud particle concentration environments, some particles may not be recorded due to the probes electronics being busy processing previous particles. These periods of probe "deadtime" are recorded for the correction of total particle concentrations (due to missed particles).

– **Three View Cloud Particle Imager (3V-CPI)**. The 3V-CPI is an inlet-based combination cloud particle probe. The probe integrates the optics and electronics of a 2D-S probe (described above) with a Cloud Particle Imager (CPI, Version 2.5) which uses a 1024 x 1024 pixel CMOS camera and data acquisition system capable of recording digital images of cloud particles with 8-bit grey scale (256 levels) at a pixel resolution of 2.3 $\mu$m and maximum frame rate of 400 frames per second. The CPI measures the size and shape of cloud particles with high resolution and enables an estimate of the relative concentration of water drops and ice particles in cloud. The CPI is triggered when both 2D-S channels detect a particle, and with appropriate depth of field corrections (e.g. Connolly et al., 2007), size distributions of ice particles greater than approximately 8 $\mu$m are obtained. Whilst the sample volume of the CPI is significantly smaller than for 2D-S, approximately 0.37 L at 100 m s$^{-1}$ airspeed, it is particularly suited to providing high resolution reference images for determining shapes and habits of ice crystals, an aid to understanding the growth history and potential origins of these particles (including identification of potential SIP mechanisms (Korolev and Leisner, 2020; Korolev et al., 2022)). Both 2D-S and CPI observe particles in the cloudy air passing down the instruments common sample tube. On occasions, these measurements can be affected by artefacts from fragmentation of particles on the inlet, so care must be taken to identify and remove these effects by various techniques (Connolly et al., 2007). This is particularly true when the inlet knife edge becomes rimed in high supercooled liquid water content conditions.

– **High Volume Precipitation Spectrometer (HVPS-3)**. The HVPS-3 (e.g. Lawson et al., 1998) uses a 128-photodiode array and electronics similar to the 2D-S probe. However, its optics are configured to provide images at 150 $\mu$m pixel resolution, giving it a nominal size range of 150-19,200 $\mu$m. This enables particles as large as 1.92 cm to be imaged, depending on the analysis technique employed. The presence of even larger particles can often be detected by observing



particle size in the direction of flight. The HVPS-3 has a typical sample volume of 310 L at an airspeed of $100\,\mathrm{m\,s^{-1}}$ and is used in this study to identify low concentrations of graupel and large precipitation particles. Data processing is similar to that of the 2D-S and further information can be found in the SPEC Inc. HVPS software manual (2010 and updates) and McFarquhar et al. (2017).

- **Holographic Cloud Probe (HALOHolo)**. An important component of the DCMEX cloud instrument suite is the HALOHolo probe. This instrument was provided by the Institute for Atmospheric Physics at the University of Mainz and Max Planck Institute for Chemistry, Mainz and is an upgraded version of the instrument described by Fugal and Shaw (2009). It consists of a 6576 x 4384 pixel detector with a pixel resolution of 2.96 $\mu$m providing a maximum 3D sample volume of 19 x 13 x 155 mm. In operation the sample volume is reduced slightly to remove potential shattering artefacts close to the optical windows leaving an effective sample volume of 35.4 cm³. HALOHolo records 3D frames at a sample rate of approximately 6 frames per second. At typical cloud penetration altitudes and air-speeds this allows a 3D volume image of cloud to be recorded every 23.3 m with a total sample volume of approximately 212.4 cm³ s⁻¹. The particle size range delivered from the 3-D volume samples is ~6 $\mu$m up to 5 mm, see Lloyd et al. (2020). Similar to the processing of the 2DS data, the distinction between frozen hydrometeors and droplets is provided by shape with the limitations as expressed in Korolev et al. (2017). A comparison between size distributions measured by HALOHolo and 2D-S, as well as the complementarity and limitations of each instrument for different habit recognition, are described in Lloyd et al. (2020) and the included citations. However, a distinct advantage of HALOHolo is that it allows a "nearest neighbour" analysis of the respective positions of cloud particles to each other within its sample volume, which is invaluable for the investigation of prospective SIP mechanisms that may have been operating in DCMEX clouds.

- **Backscatter Cloud Probe with Depolarisation (BCP-D)**. The BCP-D is a miniature backscatter cloud spectrometer based on the original Backscatter Cloud Probe (BCP) described by Beswick et al. (2014), with the addition of a polarisation detector for discriminating different cloud and dust particle phases. The BCP-D measured cloud droplet size distributions over the size range of approximately 2-50 $\mu$m which can be used to estimate total droplet number concentrations, liquid water content, median volume diameter, and effective diameter. It is normally used to detect when an aircraft encounters very large concentrations of small ice crystals (Beswick et al., 2015) and has the potential to discriminate cloud liquid and dust particles (Baumgardner et al., 2017). In this project the BCP-D is used as a backup to the more sensitive Cloud Droplet Probe (CDP).

- **Cloud Droplet Probe (CDP)**. The Droplet Measurement Technologies (DMT) CDP-2 (Lance et al., 2010) was flown on the same under-wing canister containing the BCP-D. The CDP is an open-path instrument that measures the forward-scattered light (over solid angles subtended by 1.7–14°) from the 0.658 $\mu$m incident laser beam. Particles are assigned to 1 of 30 size bins over the nominal size range 2–50 $\mu$m. Size calibration was carried out pre-flight with ten different size glass beads of certified diameter and uncertainty (Rosenberg et al., 2012). Instrument windows were cleaned before each flight and the optical alignment was found to be stable resulting in minimal changes to the calibration throughout the campaign. A campaign master calibration was obtained by taking the average of each calibration size weighted by the



uncertainty; note that data with a z-score greater than five were considered poor and discarded. The campaign calibration was applied to all flight data.. The sample area was measured at 0.262 mm$^2$ with a droplet gun during manufacturer servicing in 2021. The CDP is sensitive to large dust aerosols as well as cloud droplets. Normally conversion from scattering cross-section is done using the refractive index of water, 1.33+0i, but other refractive indicies may be applied for out-of-cloud measurements when appropriate. To obtain the highest possible spatial resolution the CDP was operated at 25 Hz.

– **Cloud Imaging Probes with resolutions 15 $\mu$m (CIP15) and 100 $\mu$m (CIP100)**. Two DMT CIPs were flown with differing resolutions. Both probes use the same 64 pixel photodiode array giving a size range of 15–930 $\mu$m and 100–6200 $\mu$m respectively (the end pixels are used for edge detection, not particle sizing). Both CIPs produce 2-bit gray-scale images which allow for more accurate small particle reconstruction (O'Shea et al., 2019, 2021). Anti-shatter tips were used on both probes.

– **Nevzorov hot-wire probe**. This probe, manufactured by Sky Physics Technology Inc., was used to measure liquid and total condensed water content (Korolev et al., 1998). From these measurements, ice water content can be calculated. The vane used, which self-aligns to the airflow, consists of two coiled wires of 2 and 3 mm diameter for LWC measurement and an 8 mm deep cup total water sensor (Korolev et al., 2013). All elements were operated at 120°C and data were recorded at 64 Hz.

– **SEA WCM-2000 hotwire probe**. This probe, described by Steen et al. (2016), has three sensing elements; liquid water content is measured with two wire elements of diameters 2.11 and 0.53 mm while the total condensed water content is measured with a concave half-pipe also of diameter 2.11 mm. Another element, oriented parallel to the airflow and free of incident water, is used to monitor changes in radiant cooling and so compensate for variations in the ambient atmospheric conditions. All elements are operated at 120°C and the sample rate was set to 10 Hz.

### 3.1.3 Wind, temperature and humidity instruments

A number of other instruments provide details of the dynamics and thermodynamics of the environment.

– **Aircraft-Integrated Meteorological Measurement System (AIMMS-20) and other wind measurements**. This instrument is manufactured by Aventech Research Inc., and was mounted in a canister under the port wing. As well as meteorological data, the AIMMS-20 measures 3D winds with a 5-port probe positioned on a 0.425 m long boom. The probe tip can be heated to inhibit ice accumulation and any water in the pressure lines can be purged with a low-pressure pneumatic system on demand. Wind data are recorded at 20 Hz with an uncertainty of 0.5 m s$^{-1}$ (Aventech Research Inc.). 3D winds are also derived from the five-hole pressure measurement system in the aircraft radome. When the aircraft penetrates supercooled cloud, ice often forms on the radome which invalidates the derived wind measurements. A small heater reduces the icing and also allows recovery from icing events. Further details are available in Petersen and Renfrew (2009) and Brown et al. (1983).





- **Airborne Vertical Atmospheric Profiling System (AVAPS) and manual dropsonde tube**. The FAAM BAe-146 is outfitted with an AVAPS (UCAR/NCAR, 1993; Hock and Franklin, 1999). Vaisala RD41 dropsondes (Vömel et al., 2021) were used throughout the campaign to obtain vertical meteorological profiles above the ground site prior to in-situ aerosol and cloud measurement runs. Before each launch, the thin-film capacitor relative humidity sensors were conditioned using the built-in AVAPS function. This provided a zero reference for the measurement (Jensen et al., 2016), resulting in an uncertainty of 2% relative humidity.

- **Aircraft-mounted video camera systems**. The aircraft has four cameras operated as standard pointing forward, back, up and down directions (relative to the airframe).

- **Lyman-Alpha evaporator probe**. Total water can be measured by this probe. Details of its specification and operation are presented by Nicholls et al. (1990) and Abel et al. (2014).

- **Humidity probes**. There were three types of hygrometers used (Price, 2022): The General Eastern 1011B and the Buck CR2 (chilled mirror hygrometers), and the Water Vapor Sensing System (WVSS-II) from SpectraSensors. A calibrated volume mixing ratio measurement is determined using the Buck CR2 and WVSS-II in combination. This setup has a response time of around 2 s. The General Eastern hygrometer acts as a backup instrument.

- **Temperature probes**. Air temperature was measured with de-iced and non-de-iced internal sensors within two Rosemount Model 102 housings (Price, 2022). These housings had similar inlets which draw flow across the sensing elements. They are designed to minimise water and particle ingress, as well as minimise interaction of the air with the walls of the inlet. As far as possible, the housings bring the air to rest relative to the aircraft. The probes used were the 17005E (loom fast probe, Non-de-iced) and 20472E (plate probe, De-Iced).

## 3.2 Langmuir Laboratory

The Langmuir Laboratory for Atmospheric Research is located near to the summit of the South Baldy Peak in the Magdalena Mountain range, the location of the DCMEX study region (Figure 1). The laboratory comprises a main building complex and separate underground (lightning protected) laboratory bunkers or "Kivas" located at the top of the South Baldy peak. Kiva-2 was instrumented with a set of aerosol, weather and electric field instruments which provided data during the field campaign.

### 3.2.1 Aerosol

Two optoelectronic aerosol spectrometers were installed in the Langmuir Kiva-2 laboratory, located on South Baldy Peak at 3,287 m above sea level, providing continuous measurements of aerosol concentrations. The first was a PLAIR Rapid-E+ bioaerosol spectrometer capable of measuring particles in the size range 0.3-100 $\mu$m using a modulated laser, linear polarizer, collection lens, and 24-pixel light detector that acquires light intensity with a high sampling rate. A second standard aerosol spectrometer, a GRIMM OPC Model 1.109, was also connected to the sample inlet and provided continuous aerosol size distribution measurements for particles from 0.25 to 32 $\mu$m in 32 size channels.



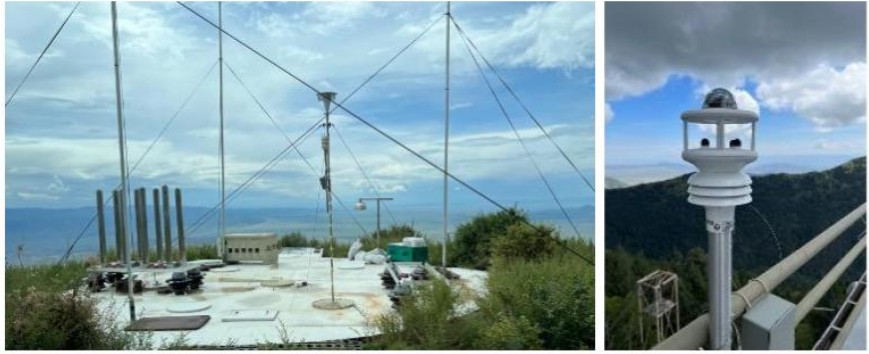

**Figure 5.** Photographs of aerosol detectors and automatic weather station locations on the Magdalena mountains during the DCMEX campaign. (Left) Kiva-2 Laboratory rooftop, South Baldy Peak, 3287 m above sea level, includes a centrally mounted University of Manchester Aerosol Inlet and Vaisalla WTX536 Met Station. (Right) Gill MaxiMet GMX600 Met Station (University of Manchester) mounted on the Langmuir laboratory rooftop railing.

In the PLAIR Rapid-E+, the time-dependent scattered light pattern generated by single aerosols are used to interpret aerosol shape and surface morphology. The Rapid-E+ determines the fluorescence properties of aerosols using a 337 UV excitation laser and measures the resultant aerosol fluorescence spectrum between 350 and 800 nm in 32 separate channels. Fluorescence lifetime was also measured in four separate wavebands 350-400, 420-460, 511-572, and 672-800 nm. The instrument is potentially capable of detecting and discriminating bacteria, fungal spores, pollen, and protein-containing dust-soil mix-

tures. Aerosols were sampled using a specialised PLAIR aerosol sampling head with improved particle sampling efficiency for aerosols up to 100 $\mu$m in diameter at a nominal flow rate of 28.3 L min$^{-1}$. The instrument was connected to a 4 m tall stainless steel sample pipe mounted to the Kiva-2 rooftop (Figure 5). The Rapid-E+ was operated remotely from the Albuquerque DCMEX operations centre via the Langmuir Laboratory internet.

    Aerosol filter sampling for INP analysis was conducted at Langmuir Laboratory using a Digitel DPA-14 programmable filter

carousel sampling system. The system was installed on the roof of the main laboratory building and sampled continuously between 15/07/22 to 28/07/22 and 05/08/22 to 08/08/22 onto polycarbonate track-etch membrane filters with 0.4 $\mu$m pore diameter (Whatman-Nuclepore 10417112) via a total suspended particles inlet achieving a typical flow rate of 20 L min$^{-1}$. Filters were pre-loaded into stacks that were programmed to change automatically at 8pm and 8am local time each day (i.e. two filters per day) in order to detect diurnal trends in INP. Analysis of INP concentrations were determined offline by DFA

and the 'wash-off' method, where the filters were placed in pure water and agitated to produce particle suspensions. DFAs on these filters were conducted in Albuquerque within 7 days of filter sampling.



### 3.2.2 Weather

A Vaisalla WTX536 meteorological station was installed at the Kiva-2 laboratory. It was placed on the aerosol sampling mast to provide collocated wind speed, direction, temperature, relative humidity, pressure, rainfall rate and hail rate.

A second meteorological station, a Gill MaxiMet GMX600 Met Station, was installed at the Langmuir Laboratory next to the Digitel aerosol filter sampler (see MaxiMet in Figure 5) providing measurements of wind speed, direction, temperature, humidity, pressure and precipitation rate.

### 3.2.3 Electric fields

Langmuir Lab maintains three "E100" electric field mills to provide storm situational awareness. They are called field "mills"
because they have a rotating blade that chops electric field (Harnwell and Van Voorhis, 1933; Malan and Schonland, 1950; Smith, 1954). The chopping allows for synchronous detection and allows accurate measurements from 1 $\mu$Hz to 10 Hz. The very low frequency measurements allow field mills to be used to observe thunderstorm charging phenomena and the transition from fair-weather to foul-weather fields. Field mills also observe the sharp transition associated with a lightning flash, but a suite of faster instruments (e.g. slow/fast antennae and RF instruments) are typically used to both locate and analyse the
lightning flash.

One field mill was mounted near Kiva2 on South Baldy peak. The E100s can measure DC electric fields up to +/- 100 kV/m. The E100 field-mills at Langmuir have been used in many studies, among the first being Standler and Winn (1979). The detailed theory of operation of the E100 used in this study is described by Winn (1984).

A slow antenna of the "LEFA" design was located on West Knoll, roughly 1.5 km Southwest of Kiva2 (Hager et al., 2012).
The LEFA is a 24-bit slow antenna capable of resolving fields from 0.01 V/m to 100 kV/m and it samples at 50 kSamples/second. LEFA can readily diagnose the difference between stepped leaders, dart leaders and K-leaders. A pair of LEFAs can differentiate intracloud for cloud-to-ground flashes.

### 3.3 Doppler radars

Two dual-polarisation Doppler weather radars were deployed during the field campaign to obtain targeted volumetric observa-
tions of the convection over the Magdelanas. One C-band dual-polarimetric Shared Mobile Atmospheric Research and Teaching (SMART) radar (SR1), (unit 1; (Biggerstaff et al., 2005, 2021)) was deployed at Socorro airport (34.022N, 106.898W) and one X-band dual polarisation solid-state radar (PX1000) (Cheong et al., 2013) was deployed at Magdalena airport (34.095N, 107.297W). Both radars operated in simultaneous transmit and receive (STaR) mode (Doviak et al., 2000). Technical descriptions of both radars are shown in Table 2, alongside a description of the WSR-88D radars at Albuquerque and Holloman (radar
IDs: KABX, KHDX) which also observe the Magdalenas with their standard, operational volume coverage patterns (NOAA, 2021).

The SMART radar collected volumes of 20 sector sweeps across a 130-degree azimuth range at elevation angles between 1.6 and 22.7 degrees followed by 5 range height indicator (RHI) scans (vertical cross section) spaced 1.5° apart in azimuth and



centered over Langmuir Laboratory. The whole volume of sector sweeps and RHIs was repeated every 5 minutes. The radar
generally came online only after deep convection had initiated.

The PX1000 radar generally came online near the beginning of the flight. Initially the radar collected volumes consisting of 20 full 360-degree PPI sweeps from 1.6 to 22.7 degrees in elevation every 5 minutes. When an echo of interest formed, the PX1000's operating mode was switched to 130 degree sectors nominally centred over Langmuir Laboratory but rotated in azimuth as needed to adequately follow the storm cell being sampled by the aircraft. The sector scans contained the same
elevation tilts as the full 360-degree volumes, but these were followed by RHI scans up to 35 or 45 degrees depending on the depth of the echo. If the storm approached the radar, a modified set of elevation tilts from 4.8 to 28.7 degrees were used to better sample the mid-to-upper portions of the cloud. Each set of tasks were repeated approximately every 5 minutes to maintain coordination with SR1.

Since the PX1000 uses a low-power solid state transmitter, pulse compression (Salazar Aquino et al., 2021) is employed
when the echoes are more than 11 km from the radar. The pulse compression led to radially-oriented artifacts that extend before and after the main precipitation feature that must be edited manually. If the target storm came closer than 10 km to the radar, a non-compressed waveform was often used. This limited the sensitivity to about 15 dBZ but removed the range artifacts.

Manual editing of the data from both radars is being performed to remove ground clutter, noise, and pulse-compression artifacts (PX1000 only) around the features that were sampled by the aircraft.

## 500  3.4  Automated cameras

Two automated cameras were developed for the campaign. Each camera instrument comprised: a Canon EOS 6D Mark II camera, a UV lens filter, a Raspberry Pi, a Mikrotik Wifi transmitter/receiver, an 8Gb SD card and a 2Tb External hard-disk. The camera had an f/1.8 50mm prime lens giving angles-of-view of 40º, 27º, 46º in the horizontal, vertical, and diagonal respectively, captured within 6240 x 4160 pixels (Canon, 2023).

The Raspberry Pi computers were running a software stack based on the camera-control software GPhoto2, with a web-based front-end written using the Python Twistd framework for control in the field. Connectivity between the two Raspberry Pis was via Secure Shell over a pair of Mikrotik wifi routers (code repository described in code availability section).

Time-lapse photographs were stored with an interval of 20 seconds. Shutter speed, aperture and ISO were automatically adjusted after every 12 photographs. For all days of camera operation there was at least one camera located at Socorro airport.
The second camera was sometimes placed at Socorro Airport, but was also tested at another location in Socorro, and also at Magdalena Airport on a number of days. Location coordinates were automatically logged in the camera metadata. Instrument scientists additionally recorded the yaw, pitch and roll of the camera set up on each day.

The timelapse images provide a useful perspective on the development of the clouds during the aircraft observations, and in addition can be used to estimate properties such as the height of cloud base and cloud top.



## 4 Case characteristics

The region around the Magdalenas Mountains in New Mexico receives the majority of its precipitation in July and August. There is substantial year-to-year variability in the amount and timing of precipitation (Prein et al., 2022). Helpfully, the majority of days within the campaign were conducive to convective cloud formation over the Magdalenas. In this section we use the extensive array of operational observation and reanalysis data to explore the general character of the meteorology, aerosol and clouds across the campaign period.

Using ECMWF ERA5 reanalysis (Hersbach et al., 2020), Figure 6 shows that as the campaign began there was low relative humidity air, with a northerly wind flow moving in on the 19th/20th July. Between the 19th and 28th July there was a transition towards a moist southerly flow with a varying easterly component at mid to upper levels. From the 28th July to the end of the campaign, mid-levels remained moist. Winds transitioned to a northerly flow around 3rd August with a westerly component at low levels, before returning to the southerly setup again before the end of the campaign.

The 700 hPa maps in Figure 6 show that the profiles over the Magdalena Mountains were part of large-scale synoptic systems. The dry northerly winds on the 19th July were associated with anti-cyclonic winds over Arizona to the west of New Mexico. The moist southerly air, present through the middle of the campaign, was part of a large-scale south-easterly flow across Mexico and Texas. The moist synoptic system described is typical of what is sometimes referred to as the North American Monsoon (Boos and Pascale, 2021).

Table 3 provides a range of statistics for each day of the campaign period. They broadly illustrate the low-level meteorological and aerosol conditions, as well as the character of the clouds that formed. The Magdalena Ridge Observatory maintains a weather station near the Langmuir Laboratory, and New Mexico Tech have shared the operational data collected during the DCMEX campaign. Table 3 includes the mean temperature and dewpoint temperature between 15-16Z (9-10am local time) from that station. This time period was chosen to represent the conditions prior to cloud formation. It is also roughly around the time the aircraft took off. The temperatures were highest when the campaign began, then dropping after the 20th July and staying fairly steady to the end of the campaign. Meanwhile, the dewpoint temperature increased after the 22nd July consistent with the increased low-level relative humidity seen in Figure 6 around the same time.

As described in Section 3.2.1, surface aerosol stations were installed for the campaign on top of the mountain. In Table 2 are the total aerosol concentration and concentration for particles larger than 2.5 $\mu$m, as measured by the ground-based GRIMM OPC. Broadly speaking, the concentration of larger aerosol particles followed the total aerosol concentration, and was only a small proportion of total aerosol (∼0.1%). Notably high aerosol days include the 23rd July, which saw the first thunderstorm of the campaign, and the 7th August, which saw one of the more intense thunderstorms during the latter portion of the campaign. Notably low aerosol days include the 31st July, which followed the day with the most intense thunderstorm and saw a later start to lightning flashes than on several of the preceeding days.

With a focus on the microphysical behaviour of the clouds, we will explore the role of cloud base temperature in influencing cloud processes. To provide an overview of cloud base temperature across the campaign, we consider an estimate of the Lifting Condensation Level temperature ($T_{LCL}$) relative to the Magdalena Observatory surface observations of temperature, dewpoint

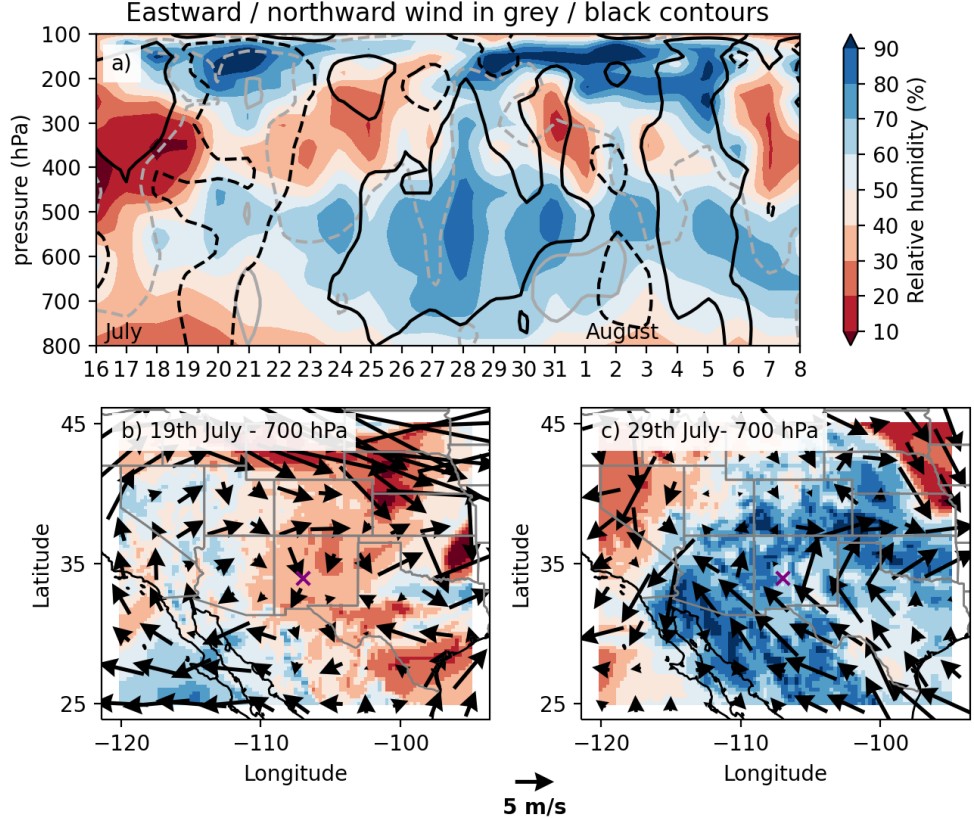

**Figure 6.** ERA5 18Z relative humidity, and zonal and meridional winds during the DCMEX campaign. a) A time-pressure plot using the mean ERA5 values over 33.5-34.5N and 106.5-107.5W (approximately the Magdalena mountains). Contour lines show 2.5 m s$^{-1}$ (solid) and -2.5 m s$^{-1}$ (dashed) winds in the northward (black) and eastward (grey) directions. In the bottom panels, the 700 hPa spatial distribution of relative humidity (filled contours, same colour scale as (a)) and wind (vectors) are shown for two illustrative days, b) the 19th and c) 29th July. Grey lines on the map show USA state boundaries and country boundaries. Black lines show coastlines. A purple cross marks the location of the Magdalena mountains

temperature and pressure. $T_{LCL}$ was calculated using the MetPy python package (May et al., 2022). For cumulus developing
into deep convection we consider the $T_{LCL}$ a reasonable approximation of the cloud base temperature.

LCL temperature remained low, and close to zero degrees, at the beginning of the campaign. It then warmed substantially to around 5-8 °C between the 23rd July and 3rd August, with exception of a dip to 2.8 °C on the 31st July. Between the 4th and 8th August the LCL temperature fluctuated with a range between 2.8 and 6.1 °C.

There is a broad relation between these cloud base temperatures and three measures of the deep convective storm charac-
teristics. Initially, we have considered the maximum deep convective cloud top height, the time of first lightning, and number





of lightning flashes. We have focused on the period 15-21Z as this was the main period of storm activity on the mountain and when aircraft flights and other observations were carried out.

Maximum cloud top heights of cloud with high optical depth (i.e. optical depth $> 23$, cloud top pressure $< 440$ hPa) ranged between 7.6 and 15.3 km. Based on this definition, the highest clouds occurred on the 26th July and the 1st and 2nd August. 560 Generally, the middle of the campaign saw higher cloud tops, consistent with these clouds electrifying. The earliest lightning flash measured by GOES GLM instrument was at 17:31Z (11:31 local) on the 28th July. This was a down-day for the aircraft. However, early lightning flashes also occured on the 25th, 27th and 30th July. With these days also having the highest number of flashes between 15-21Z.

The information in this section demonstrates that in-situ observations have been obtained for a wide range of summertime 565 convective conditions. The dataset includes days with relatively dry as well as relatively moist conditions, weakly and strongly electrified clouds, days when convection did not establish and days when convection was deep. In addition, there are a number of days with high aerosol loading and others with relatively low aerosol. As a result, a variety of case studies can be chosen depending on the scientific question of interest.

## 5    Conclusions: DCMEX aims and opportunities

The vast dataset collected during the DCMEX campaign offers widespread opportunities for scientific research for the coming decade and more. It can serve the community to inspire new discoveries, as well as assist in the development and evaluation of atmospheric models. Specifically, the variety of conditions in the large number of cases make it possible to tackle the following tasks that will contribute towards the overall aim to reduce climate sensitivity uncertainty by improving the representation of microphysical processes in global climate models.

1. Characterisation of the measured aerosol and cloud properties

  2. Evaluation of links between microphysical properties and the electrification of the clouds

  3. Analysis of ice nucleating particle measurements to establish to what extent primary ice formation explains cloud ice concentrations

  4. Characterisation of updraughts and quantification of dominant secondary ice production processes, and study of the 580      interaction of updraughts and ice production

  5. Analysis of observations and modelling of the microphysics of convective anvils, drawing on operational satellite observations

  6. Improvement of cloud microphysics schemes based on the analysis of observations

  7. Evaluation of the ability of models to reproduce the extent, thickness and lifetime of observed deep convective anvils

8. Generation of idealised warming simulations to examine the anvil response



9. Use of findings from the above activities to contribute to microphysics development within the Met Office global climate model, as both the microphysical and convective schemes are undergoing an update

By characterising aerosol and cloud properties on different days of the campaign, and by developing evidence for cloud processes, we can build a picture of variability across cumulonimbus clouds like never before. That knowledge of variability can be related to cloud radiative properties measured from satellite, as well as to local and synoptic scale weather conditions, thereby enabling a deeper understanding of potential controls on cloud radiative effect. Of course, cloud feedbacks involve variability over much longer time periods, but studying day-to-day variability can enable generation of hypotheses to be tested in longer-term datasets such as CERES satellite data combined with reanalysis products. Furthermore, by using various observations to constrain and evaluate cloud-resolving models, and in turn earth system models, we enhance these fundamental tools used to explore sensitivity of the climate to warming.

As well as mapping a route to increasing knowledge of tropical anvil cloud radiative properties and feedbacks, the above tasks also address several recommendations made by Field et al. (2017) in order to improve understanding of in-cloud microphysics:

– Improve instruments for measuring ice particles in cloud, particularly at the sub 150 micron size range

– Improve capabilities for measuring INP at T>-10°C

– Improve representation of INP in models

– Carry out integrated field campaigns that involve in-situ, remote sensing and modelling data

By enhancing knowledge and modelling of in-cloud microphysics, DCMEX activities will not only support improved understanding of cloud radiative properties but also their precipitation characteristics, another important impact of deep convective clouds. The dataset presented here is ready to use to explore a wide array of research topics. It can be used standalone to investigate hypotheses, or may form the basis of model evaluation and intercomparison.

## 6 Code and data availability

Aircraft data is available for the DCMEX flights c297-c315 at https://catalogue.ceda.ac.uk/uuid/http://catalogue.ceda.ac.uk/uuid/b1211ad185e24b488d41dd98f957506c (Facility for Airborne Atmospheric Measurements et al., 2023). Full radar data will be available at https://doi.org/10.5281/zenodo.8051426 (Carrie et al., 2023), currently SR1 data is available. Automated camera images will be available at https://catalogue.ceda.ac.uk/uuid/b839ae53abf94e23b0f61560349ccda1 (Finney, 2023). GOES data was downloaded using the goes2go python package available at https://github.com/blaylockbk/goes2go. ERA5 data was accessed through the CEDA archive (European Centre for Medium-Range Weather Forecasts, 2021). Code used for the timelapse camera connections is provided at https://bitbucket.org/ncas_it/camera/src/DCMEX-Deployment/. Code used to download GOES cloud optical depth, and then correct the latitude-longitude coordinates for parallax shift related to cloud-top height is provided at https://github.com/DLFinney/GOES_cloud_parallax_shift.





*Video supplement.* A selection of videos have been published, produced from the timelapse photography of clouds described in Section 3.4. These are available to download from Finney et al. (2023).

*Author contributions.* DLF and AMB led the writing of the manuscript. AMB is principle investigator of the DCMEX project and led the field campaign with the help of many of the co-authors, and others at FAAM. DLF provided analysis and figure production. MG, HW,
GN, MB, RGS, MD, DW and DD provided descriptions of instrument operations. HW, GN, DD and JC also provided analysis and/or figure production. KB, SB, TC, JC and JG quality reviewed the manuscript text at key stages of drafting. PRF, HC, BJM, GL, NAM, MF, KH, NMT, PIW, JR, GC, RM, GA, RRB and PJC were fundamental to collection of data for the project and contributed to the writing and reviewing of the manuscript.

*Competing interests.* The authors declare that they have no conflict of interest.

*Acknowledgements.* This work was supported by the Natural Environmental Research Council (NERC; NE/T006420/1 and NE/T006439/1). Airborne data was obtained using the BAe-146-301 Atmospheric Research Aircraft [ARA] flown by Airtask Ltd and managed by FAAM Airborne Laboratory, jointly operated by UKRI and the University of Leeds. We acknowledge the NCAS Atmospheric Measurement and Observation Facility (AMOF) a UKRI-NERC funded facility, for providing the 3V-CPI and HVPS-3 cloud probes and Aerosol Mass Spectrometer. We are very grateful to all of the people from FAAM, Airtask and Avalon that worked so hard to make the DCMEX flying campaign
possible. In particular we would like to thank the pilots (Steve James, Sean Finbarre Brennan and Mark Robinson) for innovative and careful flying in difficult conditions, and Doug Anderson (FAAM) for dealing so well and pleasantly with all the logistics. And to Fran Morris (University of Leeds PhD) for her diligent support of Doug in the early campaign logistics. Thank you in addition to the other University of Leeds PhD students who helped during the campaign: Greg Dritschel, James Bassford and Kasia Nowakowska. Thanks to Marc Gross, David Bennecke, Luis Contreras, Vicki Kelsey and Stetson Reger, students at New Mexico Tech (NMT) for their help with lightning warn-
ings and other aspects of the project. Thanks to Bill and Eileen Ryan from NMT for providing the Magdalena Observatory weather station data operationally collected during the campaign. Thanks also to Harald Edens and Ken Eack, previously at NMT and now at Los Alamos, for being so enthusiastic about hosting DCMEX at Langmuir Laboratory and for so much help in the early planning stages.



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



**Table 1.** Overview of flights and their sampling features. Aerosol run heights are mean height above sea level (ASL) from the GIN altitude instrument, asterisks mark runs that were terrain following. Many of the cloud runs are comprised of grouped individual cloud passes that are separated by less than 60 sec. Only runs lasting longer than 5 sec and with altitudes above 4km are counted. Near-cloud temperature for the lowest and highest altitude cloud passes were averaged from the 1 Hz measurements in the 15 seconds before entering cloud. The deiced temperature was used for temperatures <=273 K, and the non-deiced temperature used for >273K. If the preceeding 15 seconds contained no data, then the post-cloud 15 second period was used, if data were available.

| Date | ID | Take-off and landing time (UTC) | Aerosol run heights (km ASL) | Cloud runs (number and near-cloud T range) | Notes |
|------|----|--------------------------------|------------------------------|--------------------------------------------|-------|
| Sat 16 Jul | C297 | 16:10 - 19:07 | 2.3*, 2.5, 2.6, 4.8 | 3 (274 to 273) | Test flight |
| Tue 19 Jul | C298 | 15:40 - 19:55 | 2.3*, 4.8 | 23 (275 to n/a) | Outflow sampled |
| Wed 20 Jul | C299 | 16:14 - 20:08 | 2.2*, 4.8 | 24 (280 to 247) | – |
| Fri 22 Jul | C300 | 15:40 - 20:04 | 2.3*, 4.8 | 31 (278 to 250) | – |
| Sat 23 Jul | C301 | 15:27 - 19:58 | 2.2*, 5.1, 6.0 | 26 (279 to 248) | Cell electrified / Outflow sampled |
| Sun 24 Jul | C302 | 15:29 - 19:04 | 2.5*, 4.5, 4.6 | 10 (n/a*) | Overcast, no convection / Aborted flight early |
| Mon 25 Jul | C303 | 15:30 - 19:55 | 3.5, 4.6, 6.5 | 26 (276 to 252) | 2 cells electrified / Outflow sampled |
| Tue 26 Jul | C304 | 15:01 - 19:31 | 2.5*, 4.5, 5.8 | 29 (277 to n/a) | Cell electrified |
| Wed 27 Jul | C305 | 15:36 - 20:05 | 3.2, 3.5, 6.5 | 24 (278 to n/a) | 1 cell electrified / Cloud base sampled |
| Fri 29 Jul | C306 | 15:27 - 19:54 | 2.1*, 5.4 | 27 (276 to 255) | – |
| Sat 30 Jul | C307 | 15:24 - 19:54 | 2.1*, 2.8, 6.7 | 16 (276 to 260) | 2 cells electrified |
| Sun 31 Jul | C308 | 15:30 - 20:04 | 2.1*, 5.1, 7.3 | 28 (276 to 245) | 2 cells electrified / Outflow sampled |
| Mon 1 Aug | C309 | 15:43 - 20:07 | 2.1*, 5.4, 6.7 | 26 (278 to 263) | 1 cell electrified / Stratiform sampled |
| Tue 2 Aug | C310 | 15:26 - 20:00 | 2.0*, 2.1*, 4.5, 7.1 | 18 (280 to 253) | Sampled cloud street in valley / Clouds electrified |
| Wed 3 Aug | C311 | 15:26 - 18:14 | 1.9*, 2.1*, 5.1, 6.5 | 6 (273 to 258) | No convective cloud / Aborted flight early |
| Thu 4 Aug | C312 | 16:05 - 20:37 | 2.1*, 4.4, 6.5 | 31 (278 to 263) | – |
| Sat 6 Aug | C313 | 15:26 - 19:35 | 1.9*, 2.1*, 4.5, 6.5 | 21 (278 to 266) | – |
| Sun 7 Aug | C314 | 15:57 - 20:01 | 2.1*, 6.7 | 27 (279 to 256) | 1 cell showed high reflectivities |
| Mon 8 Aug | C315 | 15:57 - 19:15 | 4.4 | 33 (275 to 262) | 1 cell had high reflectivity / Extensive sampling at -5°C |

* excluded due to highly varying altitude during long stratus cloud passes



**Table 2.** Technical specification of radar instruments.

|  | SMART | PX1000 | WSR-88D |
|---|---|---|---|
| Frequency band | C-band | X-band | S-band |
| Beamwidth (°) | 1.5 | 1.8 | 0.9 |
| Transmitter | Magnetron | Solid-state | Klystron |
| Transmit power (kW) | 250 | 0.1 | 750 |
| Range resolution (m) | 150 | 60 | 250 |
| Azimuthal resolution (°) | 1.0 | 1.0 | 0.5 |
| Distance to Langmuir Laboratory (km) | 27 | 17 | 130 / 160 |
| Sector range | Variable | Variable | 0-360 |
| RHI range (km) | 120 | 60 | N/A |



**Table 3.** Ground-based aerosol and weather measurements, and satellite estimates of cloud top height and lightning. Aerosol is obtained by the GRIMM instrument located at Langmuir laboratory. Temperature (T) and Dew point temperature ($T_d$) are obtained from the operational weather station at the Magdalena Ridge Observatory. Temperature at the Lifting Condensation Level (LCL) is estimated from the temperature and dewpoint. All ground based measurements and estimates are averaged over the hour 15-16Z to represent conditions prior to convection. Satellite data is processed for the 15-21Z, 6-hour period to roughly represent the flight period. Estimates of cloud top height are taken as the maximum GOES value within a rectangular region with edges passing through the points of the kite in Figure 1, based on 5 minute images when available. Only clouds with an optical depth $> 23$ and cloud-top pressure $< 440$ hPa are considered, consistent with the ISCCP definition of deep convective cloud. The GOES cloud fields were corrected for parallax shift as described in Figure 1, and regridded to a $0.01°$ lat-lon grid using nearest neighbour method. Lightning flashes are counted from the GOES GLM instrument within a rectangular box whose corners are the mid-points of the kite edges in Figure 1. The number of flashes within 15-21Z as well as the time of first flash are given.

| Date | Ground (15-16Z) aerosol total $L^{-1}$ | aerosol ($> 2.5\mu m$) $L^{-1}$ | T °C | $T_d$ °C | $T_{LCL}$ °C | Satellite (15-21Z) Cloud top max km | Lightning # / UTC |
|---|---|---|---|---|---|---|---|
| 16 Jul* | 15600 | 2 | 17.0 | 5.5 | 3.0 | – | 0 |
| 17 Jul | 44900 | 21 | 18.0 | 5.3 | 2.6 | – | 0 |
| 18 Jul | 18900 | 11 | 17.8 | 2.8 | -0.3 | – | 0 |
| 19 Jul* | 16900 | 18 | 17.9 | 3.3 | 0.3 | 7.6 | 0 |
| 20 Jul* | 18300 | 14 | 17.9 | 4.3 | 1.5 | 12.7 | 0 |
| 21 Jul | 12200 | 18 | 15.4 | 4.5 | 2.1 | 12.4 | 0 |
| 22 Jul* | 20700 | 13 | 17.8 | 5.3 | 2.7 | 11.8 | 0 |
| 23 Jul* | 52300 | 42 | 14.7 | 6.8 | 5.1 | 10.5 | 3 (19:14) |
| 24 Jul* | 23500 | 4 | 13.1 | 6.8 | 5.4 | 11.0 | 0 |
| 25 Jul* | 42600 | 28 | 13.8 | 8.7 | 7.6 | 11.8 | 34 (17:49) |
| 26 Jul* | 30200 | 4 | 12.9 | 8.1 | 7.1 | 14.8 | 13 (19:38) |
| 27 Jul* | 16200 | 4 | 14.0 | 7.9 | 6.6 | 13.3 | 44 (16:50) |
| 28 Jul | 22900 | 11 | 12.8 | 7.7 | 6.5 | 12.9 | 36 (17:31) |
| 29 Jul* | 24000 | 18 | 13.3 | 7.3 | 6.0 | 11.2 | 2 (19:46) |
| 30 Jul* | 14800 | 7 | 12.5 | 8.3 | 7.4 | 13.7 | 46 (17:37) |
| 31 Jul* | 7510 | 2 | 13.3 | 4.6 | 2.8 | 11.9 | 29 (18:51) |
| 1 Aug* | 13300 | 4 | 14.0 | 6.8 | 5.3 | 14.4 | 1 (19:45) |
| 2 Aug* | 10300 | 4 | 14.5 | 6.7 | 5.0 | 15.3 | 15 (19:27) |
| 3 Aug* | 18200 | 9 | 12.3 | 7.4 | 6.3 | 10.9 | 0 |
| 4 Aug* | 28400 | 4 | 14.9 | 6.5 | 4.6 | 12.2 | 0 |
| 5 Aug | 12400 | 2 | 14.1 | 5.3 | 3.4 | 12.5 | 0 |
| 6 Aug* | 40700 | 31 | 14.9 | 6.9 | 5.2 | 11.0 | 7 (18:40) |
| 7 Aug* | 59000 | 72 | 13.2 | 4.6 | 2.8 | 11.6 | 24 (18:20) |
| 8 Aug* | 24300 | 12 | 14.0 | 7.5 | 6.1 | 9.7 | 0 |

* Flight day