# Peer review of "DCMEX coordinated aircraft and ground observations: Microphysics, aerosol and dynamics during cumulonimbus development"

_Earth System Science Data, 2023_

## Author Comment (AC1)

*Line numbers refer to the marked-up document.*

Response to reviewer comment RC1

This manuscript offers an extensive compilation of data for cloud microphysics research. My own research focuses on the transformation of aerosols in the atmosphere, so I am more concerned about aerosol measurements on aircraft. I believe that the datasets from aerosol instruments have covered virtually all the key aspects of aerosol characteristics necessary for investigating aerosol-cloud interactions. The only concern to me is the sampling rates of these individual instruments and the methods to synchronize their measurements. It would help a lot if the authors could provide a summarizing figure (similar to Figure 4) showing the range of aerosol sizes and the corresponding sampling rates for aerosol measurements.

*We thank the reviewer for these helpful comments and for highlighting the additional information we can provide to improve the useability of the data.*

*We have added the following text on instrument synchronisation:*
*"All instruments in this dataset were time synchronised with the FAAM on-board time server. Two Meinberg LANTIME M600/GPS/PTP Stratum 1 time servers on board provide Precise Time Protocol (PTP) Version 2 and Network Time Protocol (NTP) reference time signals to all PTP and NTP compatible systems connected to the aircraft network. They are updated to Institute of Electrical and Electronics Engineers (IEEE) 1588-2019 standard with one being configured as the Grandmaster Clock so that all PTP clients use the same server. The second M600 is there for redundancy and will switch from passive to Grandmaster when required. All measurements should thus be synchronised to the same time stamp on a microsecond (for PTP) or millisecond (NTP) scale." L166-172*

*The exception was the HALOHolo instrument, but this instrument data will not be published along with this dataset. We nevertheless choose to mention its collection, and at that point in the text we include " HaloHOLO was the only instrument not time synchronised during flight. Instead, it was time synchronised in post processing by correlating its in-cannister ambient pressure data with core FAAM pressure data." L603*

*Figure 4 has had aerosol instruments added which shows the sampling rates for aerosol instruments. This inclusion has made the previous figure 3 largely redundant so we have removed that.*

[Figure]

Response to reviewer comment RC2

The authors describe aircraft and ground observations during the DCMEX campaign in 2022. The article contains information about the campaign operations and the instrumentation on the ground and deployed on the FAAM BAe-146 aircraft.
The manuscript and the data published along with it fit the scope of ESSD very well. The data set presents a unique set of airborne and ground based observations microphysics of deep convective clouds. This has the potential to provide valuable bases for future analysis. However, after reviewing both the manuscript and the data, the aim of this publication is not entirely clear to me. As a general remark: I am reviewing the manuscript and the data from the standpoint of a potential user who wasn't part of the project and would like to do some analyses with these data sets.
The paper and the published data read to me as a loose collection of information about the campaign. There is no clear structure and the published data sets don't appear to be especially treated for this publication and for future users. The data sets all have some standard format without much documentation or description on what to find where, version information, development steps, quality control measures. I would expect the data sets that are published along with an ESSD publication to be more adapted and user friendly as is currently the case.

I would recommend publication only after a major restructuring of the paper and substantial improvements on the presentation and documentation of the data sets.

We thank the reviewer for taking the time to consider our manuscript and data. We are pleased that the reviewer considers the manuscript and dataset to fit well with the ESSD journal.

We have addressed their comments below  and hope that they will now find the dataset more approachable. In summary: we have refocused the abstract and summary section to provide a clearer purpose for the paper; we have published data from many more

instruments; we have now clearly separated unpublished instruments in the text and greatly reduced their description; and we have included a section which describes dataset structure, content, key files, and their naming. We believe this paper now offers access to the dataset for a wider audience than the project team.

**General remarks about the manuscript:**

The aim of this paper is not clear. I think the paper suffers from too many points of view that the authors tried to include in this paper. Some examples:

- The abstract mentions the relation of the DCMEX project to other research projects. This might not be necessary to understand the content of the paper. Or, at least, this doesn't need to be part of the abstract as this is not really a topic of the rest of the paper.1

Having revisited the abstract, we agree with the reviewer that it could be more focused on the content of the paper, and less on the wider context of the observations. We have removed the research programme reference from abstract. Indeed, we have reworded the whole abstract to focus more on the content of the paper.

"Cloud feedbacks associated with deep convective anvils remain highly uncertain. In part, this uncertainty arises from a lack of understanding of how microphysical processes influence cloud radiative effect. In particular, climate models have a poor representation of microphysics processes, thereby encouraging collection and study of observation data to enable better representation of these processes in models. As such, the Deep Convective Microphysics Experiment (DCMEX) undertook an in-situ aircraft and ground-based measurement campaign of New Mexico deep convective clouds during July-August 2022. The campaign coordinated a broad range of instrumentation measuring aerosol, cloud physics, radar, thermodynamics, dynamics, electric fields and weather. This paper introduces the potential data user to DCMEX observational campaign characteristics, relevant instrument details, and references for more detailed instrument descriptions. Also included is information on the structure and important files in the dataset in order to aid accessibility of the dataset to new users. Our overview of the campaign cases illustrates the complementary operational observations available, as well as demonstrating the breadth of the campaign cases observed. During the campaign, a wide selection of environmental conditions occurred, ranging from dry, northerly air masses with low wind shear, to moist, southerly air masses with high wind shear. This provided a wide range of different convective growth situations. Of 19 flight days only 2 days lacked formation of convective cloud. The dataset presented will help establish new understanding of processes on the smallest, cloud and aerosol particle scales and, once combined with operational satellite observations and modelling, can support efforts to reduce uncertainty of anvil cloud radiative impacts on climate scales" L1-26

- lines 115 to 126: a lot of details about the flight operations are given, without these

playing another role in the rest of the manuscript. These kinds of decisions are part of every aircraft campaign. Why is this level of detail necessary for future data users?

Having revisited this text with the reviewer's comments in mind, we agree that this paragraph is superfluous and have removed it.

- Section 3 (Instrumentation): this is a very long section with a lot of details on the measurement principles of all instruments deployed during the campaign. Especially, subsections 3.1.1, 3.1.2, and 3.1.3 seem unnecessarily detailed. Given that a lot of instrument sections contain references to previous publications, these descriptions can be shortened with referencing papers that describe the instruments in more detail to help readability of this manuscript. In addition, only data from a fraction of the instruments listed in this paper is actually published along with it (see comments below). This left me confused about the goal of this instrument list.

We appreciate there is a lot of information in these sections, we have read through and made some parts more concise. We have published data from many more instruments and this section now only includes those instruments. We have kept the majority of text because we feel that it will be important to some users, though we accept it may seem too verbose to others.

- The concluding chapter doesn't really conclude the paper but opens up another topic with listing a lot of potential future analysis opportunities. Which could definitely be part of a concluding chapter. However, without any real conclusion, the reader is left wondering, what paper was all about.

We note some previous ESSD publications (e.g. https://doi.org/10.5194/essd-15-5785-2023) have used a "Summary" as the final section. Having considered the reviewer's comment, we feel like a Summary section would better suit our paper. We have reworded the final section accordingly.
"The DCMEX campaign has collected a wide range of observation data of convective cloud growth in New Mexico over the period July-August 2022. Collected data included measurement of aerosol, cloud physics, radar, thermodynamic and dynamic variables. In addition, a collection of timelapse imagery of the cloud growth was obtained.

The study was focused over the Magdalena mountains where reliable orographic convection occurs during the summer. Convective cloud growth was observed on 17 of the 19 flight days. Day to day environmental conditions varied in terms of source air mass, humidity, and wind shear. As a result, the dataset includes convective cloud forming at a range of speeds and intensities. The range of data allows analysis of primary and secondary ice formation under different conditions and, when combined with modelling and operational satellite data, the dataset enables analysis of the influence of microphysical processes on cloud radiative effect.

This paper has introduced the necessary details of the campaign and dataset to enable researchers external to the project to use the DCMEX observation data. The dataset

offers opportunities to understand aerosol-cloud interactions, cloud physics and can be used with modelling and operational data to understand cloud radiative effects"
L753-800

**Remarks about the aircraft data sets:**

- Only data from a small fraction of the instruments that were listed in the manuscript are actually published. There isn't explained anywhere. After reading the paper, I would have expected data from all instruments to be published as assets to the paper.

We have now published much more data, and clearly segregated out mention of any unpublished instrument data. We have kept a list of instruments that are not yet published because we feel they are important for a full description of the campaign, and because it allows other scientists to know they can approach the team for collaboration in advance of them separately being published.

- No overview was given in the text which kind of files are uploaded or how to navigate the folder structure.

There is a new section, "Dataset archive details", which outlines the key files and directory structure of the data.

- What do different versions or r mean? Where is the version history so that users can assess which version they would like to use and what was changed between the versions?

The new section highlights that v and r are version and revision numbers in filenames. The revision number changes are described in the netcdf metadata under the global attribute, "revision_comment". A reference for version number has now been provided in the text, https://zenodo.org/records/8082628

- The bulk download of data in the CERES website is not working. However, this might have more to do with the data base and something the authors can't control.

We think the reviewer means "CEDA". Both CEDA and CERES download facilities are not under our control.

- Where is data from flight c315? And in turn, why is flight c296 part of the DCMX data record?

C315 was uploaded but it had not been listed with the collection. That has now been corrected. Thank you for spotting it.

C296 was the UK test flight, so whilst it is archived with the dataset, it is not addressing the aims of the dataset. The flight is now referred to in the text:

"Basic details regarding the cloud and aerosol runs are provided in Table 1. In addition to the flights listed here, there is a UK test flight included in the dataset with flight ID, c296." L104

- There are a lot of files in archive without any explanation about their generation and what they contain. For example, what is flight-cst_faam_20220806_r0_c313.yaml? What is core_faam_20220730_v005_r0_c307.nc?

The new section, "Dataset archive details", now highlights the important files for new users and explains the naming conventions.

**Remarks about the radar data sets:**

- The tar.gz files have too many unnecessary subfolders (DCMEX/SR1/data//files...).

We have removed the unnecessary "moment" folder in version 3 of the dataset archived on Zenodo but believe the rest of the file structure is meaningful.

- The tar.gz files are huge, not everyone is interested in all 41 different versions. Those should be split up into either different files. And again, no explanation is given about the differences in these versions.

It is unclear what the reviewer is referring to by "versions". There are links for the operator logs for each instrument for each event and separate links for each instrument for each day. The text in the Zenodo site states what the links represent.

We have also added some additional text to the paper to explain filenaming:
"Radar data are archived at Carrie et al. (2024). The files from each day of operation are zipped into an archive file. Within those files, each individual radar sweep (sector or Range-Height Indicator (RHI)) are stored with the following naming convention: cfrad.<start day>_<start time>_to_<end day>_<end time>_<radar name>_v<N>_s<n>_<el / az>_<PPI or RHI>.nc. Start day/end day is in the format YYYYMMDD and start time/end time is in the format HHmmss.fractionalsecond, N is the volume number through the day (consecutive sweeps or RHIs are grouped into a contiguous volume), n is the number of the sweep within the volume, el or az is the fixed elevation angle of the PPI or fixed azimuth angle of the RHI respectively, and PPI550 or RHI denotes the orientation of the scan. Each netcdf file contains the radar location along with parameters for that particular scan within the metadata as per the cf-radial file convention (NCAR, 2016)" L690-697

We believe that most users will be interested in the data for a particular event for a given instrument. Hence, the tar files are organised in that manner. Radar data sets are generally large. We believe the tar files are typical of other radar dataset archives and,

hence, did not break the files down further. Moreover, zenodo has a 100-file limit. Breaking the files down further would require more than one zenodo doi reference. We did add text on the zenodo site to indicate the general size of the uncompressed files so users will know to use appropriate computing resources.

- The radar data files contain a lot of calibration data which is probably unnecessary for most users. And without any explanation either in a publication or in the netcdf files, it is hard to judge which variables one needs. I would suggest to either remove the bulk of the calibration constants, etc. or to substantially improve the description in the netcdf attributes of these files.

The data format is the current standard in the weather radar research community. Given the need to satisfy a diverse suite of research instruments, the format contains many fields that often don't apply to a specific instrument. The redundant or apparently unnecessary metadata is often still required by radar data processing software that is in use by the research community. If we removed fields, it is possible that the data would no longer be accessible using these standard processing software tools.

- Data from the time periods 20220720-20220727 and 20220803-20220804 are missing without an explanation.

The latest version of the zenodo archive contains files for the PX1000 radar that was operating during those days in which flights occurred. The SR1 data had several periods of down time due to antenna control failures. Text has been added to the zenodo site to acknowledge that SR1 was nonfunctional during several of the flight days.

- There are a lot of unnecessary entries for a ground based radar (altitude_agl only nan, eastward velocity, ...).

The current recognized standard format for radar data attempts to satisfy all platforms in the research community. The radar processing software available and in use within the research community expects those fields to exist, even if they are unused. Hence, these fields must be kept. We do recognize the challenges in developing user-specific software to process these data.

- This data set is not designed for the end user but for the radar specialist. In the current state, the published data is not usable for someone not part of the project or who is well versed in radar data analysis. I would expect a data set that is published in ESSD, to be in a usable format for a wider community. This is not the case for the radar data right now. The data would need major restructuring along with a more detailed explanation of the published data in the manuscript.

We concur that the research radar data is archived in a manner consistent with standards used in the research radar community. To that extent, the reviewer is correct that the archive is designed for radar specialists. However, the data format is netCDF and well documented. Hence, general netCDF format tools are available to access the

data using non-specialists' software.

- SR1 data is incomplete, PX1000 not published at all.

The latest version of the data on the zenodo site has the complete SR1 data set, which has been edited for the flight days. We also include the raw PX1000 data. However, there are problems with the elevation angle metadata for some of the PX1000 volumes that has just recently come to light. We are working to correct these files and will make them available as soon as possible as another version of the data on the zenodo site. Text has been added to the zenodo site detailing which dates and times have incorrect recorded elevation angles in the PX1000 data.

**General comments about the manuscript:**

- Unit formats in the text are all over the place. Some are missing spaces between number and unit, sometimes exponents are used, sometimes a '/' in text for the same unit. Sometimes they are written in a formula environment, resulting in a different font.

We have made these features consistent.

- The text mentions a lot of instruments being operated together on the aircraft. Was there any sort of time correction and correlations done to ensure simultaneous observations?

*We have added the following text on instrument syncrhonisation:*
*"All instruments in this dataset were time synchronised with the FAAM on-board time server. Two Meinberg LANTIME M600/GPS/PTP Stratum 1 time servers on board provide Precise Time Protocol (PTP) Version 2 and Network Time Protocol (NTP) reference time signals to all PTP and NTP compatible systems connected to the aircraft network. They are updated to Institute of Electrical and Electronics Engineers (IEEE) 1588-2019 standard with one being configured as the Grandmaster Clock so that all PTP clients use the same server. The second M600 is there for redundancy and will switch from passive to Grandmaster when required. All measurements should thus be synchronised to the same time stamp on a microsecond (for PTP) or millisecond (NTP) scale." L166-172*

The exception was the HaloHOLO but this is noted in the instrument's short description in the revised paper.

**Specific comments on the manuscript:**

- Figure 2: what do the dashed and solid lines depict?
The following text has been added to the caption:
"Dashed lines indicate when the instruments were operational, solid lines indicate representative periods when instruments were able to detect the cloud." L152

- Figure 3: font size too small

With adjustments to figure 4 following reviewer 1's comments, Figure 3 has become redundant and been removed.

- l 172: parentheses missing around citation Drewnick et al. (2005)

Thank you, corrected.

- l 278 and l 282: how was "in cloud" and "out of clouds" determined? Which quantities where used to determine this, as there is usually a gradual transition between free air and clouds.

The following text has been added:
"During campaign flights, it was necessary to determine if the upcoming run was a cloud run in order to set the appropriate operation of the CVI inlet. The cockpit crew would announce cloud runs prior to entering cloud based on line-of-sight. For these in-clouds runs, cloud residues were sample…" L316-318

- Figure 4: unit on the ordinate should be in proper SI format

This has been changed to L s$^{-1}$

- l 383 and 388: "hot-wire" and "hotwire"

Corrected to "hot-wire"

- Sec 3.1.3 title: this section also contains camera images from the aircraft and not only wind, temperature and humidity.

Changed to "Wind, temperature, humidity and imagery instruments"

- Sec 3.3: would be great to mention that all three radars operate on different wave lengths and thus observe different drop sizes.

All radar wavelengths observe the same drops but the scattering does vary depending on drop size and wavelength. We have added the following text.

Added; : "Given the differing wavelengths of the radars, they exhibit varying interaction with hydrometeors, particularly those of larger diameters." L551

- l 559: are the heights given above ground or above MSL?

All heights given in the paper are above sea level (ASL). We have mentioned this at first use (L96) and included "ASL" throughout.

- l 607: link to data set is incorrect

Thank you for pointing this out. Our apologies. The asset link was correct but a typo must have crept into the tex file. It is now corrected to https://catalogue.ceda.ac.uk/uuid/b1211ad185e24b488d41dd98f957506c

- l 611: why is the code for goes2go mentioned in the data and code section? I did not have the impression that this was part of the paper. If it is merely a code that was used, it should be properly cited but not mentioned in this section.

The satellite imagery was used in figure 1c and for cloud top height and lightning analysis in table 3. We have removed the text from the code availability, and instead cited it in the text.

- Why is the repository (https://bitbucket.org/ncas_it/camera/src/DCMEX-Deployment) listed here? See my comment above (l 611). If the code was developed as part of this manuscript, it should be explained properly. If this is code that was used, please cite it as code.

We have removed the text from the code availability, and instead cited it in the text.

- l 615: why is the code for the satellite pictures mentioned if they are not part of the paper? The data is not published either.

The satellite imagery was used in figure 1c and for cloud top height and lightning analysis in table 3. We have removed the text from the code availability, and instead cited it in the text.

---

## Author Response (AR2)

*We thank the editor and review for their time. Here are responses to the most recent comments/questions:*

Figure 1: description line 6: change C) to c)

*Changed*

Line 396: Vaisalla -> Vaisala; and WTX -> WXT

*Changed*

Line 582: latter -> later

*Changed*

What are the data collection frequencies of different instruments and how do you suggest aligning the data points? For example, some particle size distribution measurements may take 1 min, others may just 1 s, how will you use these data?

*The published frequency of each dataset is inferrable from the metadata. When combining data on different frequencies an averaging approach is likely needed. However, the exact method may depend on the purpose of the analysis and the combination of variables. We cannot anticipate all potential approaches but are happy to work with anyone who wishes to discuss with us the best way of processing the data for their needs.*

*Specifically in relation to aerosol particle size distributions we can provide the following guidance. When calculating the mass concentrations, we average SP2 data at 1 Hz onto the AMS time base (for time base, refer to file metadata description or the start/end run time for each run). For SMPS, a size distribution is recorded every ~1 minute. As this is a coarse resolution, we suggest only using the SMPS data during straight and level runs on the flight track.*